# Fight the Cancer, Hit the CAF!

**DOI:** 10.3390/cancers14153570

**Published:** 2022-07-22

**Authors:** Andrea Papait, Jacopo Romoli, Francesca Romana Stefani, Paola Chiodelli, Maria Cristina Montresor, Lorenzo Agoni, Antonietta Rosa Silini, Ornella Parolini

**Affiliations:** 1Department of Life Science and Public Health, Università Cattolica del Sacro Cuore, 00168 Rome, Italy; andrea.papait@unicatt.it (A.P.); jacopo.romoli@unicatt.it (J.R.); 2Fondazione Policlinico Universitario “Agostino Gemelli” IRCCS, 00168 Rome, Italy; 3Centro di Ricerca E. Menni, Fondazione Poliambulanza Istituto Ospedaliero, 25124 Brescia, Italy; francesca.stefani@poliambulanza.it (F.R.S.); paola.chiodelli@poliambulanza.it (P.C.); antonietta.silini@poliambulanza.it (A.R.S.); 4Pathological Anatomy Unit, Fondazione Poliambulanza Istituto Ospedaliero, 25124 Brescia, Italy; cristina.montresor@poliambulanza.it; 5Obstetrics and Gynecology Unit, Fondazione Poliambulanza Istituto Ospedaliero, 25124 Brescia, Italy; lorenzo.agoni@poliambulanza.it

**Keywords:** cancer-associated fibroblasts (CAF), stroma, tumor microenvironment (TME), preclinical studies, clinical trials

## Abstract

**Simple Summary:**

In the last 20 years, the tumor microenvironment (TME) has raised an increasing interest from the therapeutic point of view. Indeed, different strategies targeting either the endothelial or the immune component have been implemented. Furthermore, cancer-associated fibroblasts (CAF) have attracted even more interest due to their ability to prime the TME in order to favor tumor progression and metastasis. This current review provides a comprehensive overview on the latest discoveries regarding CAF, more specifically on their complex characterization and on preclinical studies and clinical trials that target CAF within the TME.

**Abstract:**

The tumor microenvironment (TME) is comprised of different cellular components, such as immune and stromal cells, which co-operate in unison to promote tumor progression and metastasis. In the last decade, there has been an increasing focus on one specific component of the TME, the stromal component, often referred to as Cancer-Associated Fibroblasts (CAF). CAF modulate the immune response and alter the composition of the extracellular matrix with a decisive impact on the response to immunotherapies and conventional chemotherapy. The most recent publications based on single-cell analysis have underlined CAF heterogeneity and the unique plasticity that strongly impact the TME. In this review, we focus not only on the characterization of CAF based on the most recent findings, but also on their impact on the immune system. We also discuss clinical trials and preclinical studies where targeting CAF revealed controversial results. Therefore, future efforts should focus on understanding the functional properties of individual subtypes of CAF, taking into consideration the peculiarities of each pathological context.

## 1. Introduction

### The Tumor Microenvironment: A Focus on Stromal Cells

Tumor cells are only one of the many partners involved in tumor development and progression. Indeed, the importance of the tumor microenvironment (TME), referred to as the cells and interactions that exist around the tumor cells that play a key role in supporting tumor growth, is now widely acknowledged. This concept initially started in 1889 with the seed and soil theory, where the English surgeon Stephen Paget, in “The distribution of secondary growths in cancer of the breast”, proposed the existence of a “congenial soil” that nourishes and fosters growth and survival of the metastatic tumor [1]. Yet, it was only in the early 1970s that pioneer researchers with an integrated view of the tumor, including immunity and angiogenesis, published their work. The first immunological studies reported the capacity of T lymphocytes, natural killer (NK) cells and macrophages to infiltrate the tumor mass [2,3,4,5,6,7]; as well, the presence of immunoglobulin and complement proteins was observed in the TME [8,9]. Judah Folkman, among others, discovered the process of angiogenesis in cancer [10,11] and a dependency between tumor growth and neo-angiogenesis was suggested [11,12].

In the last two decades, however, the focus has been restructured to include other TME components. The cells present in the TME can be roughly categorized into two types, those that are present in the parenchyma of healthy tissue before the tumor develops, and those that are recruited by the tumor itself. Indeed, the source of stromal cells in the TME is still debated and a large spectrum of precursors has been proposed [13,14].

In some tumors, such as pancreatic cancer, there is evidence of stromal cells that derive from either bone marrow–mesenchymal stromal cell (BM–MSC) [15,16,17], or from endothelial cells [18]. Studies performed in other cancer models suggest that the stromal component present in the TME may also result from the differentiation of resident fibroblasts [19], adipocytes [20], adipose-derived MSC (AD-MSC) [21,22], hematopoietic stem cells (HSC) [23,24], pericytes [25,26], and epithelial cells undergoing epithelial-to-mesenchymal transition (EMT) [27,28], as shown in Figure 1A. Overall, all of these cell types become part of the TME and are influenced by the tumor cells in a dualism where both influence each other, contributing to build what is the TME. Indeed, stromal cells can either exert an antitumor effect through the release of specific factors, such as transforming growth factor beta (TGF-β) [29], or they can prime the TME by promoting tumor growth through immune evasion, stimulating neo-angiogenesis, as well as also supporting the process of tumor proliferation and metastasis. This tumor-supporting stromal component is nowadays mostly referred to as cancer-associated fibroblasts (CAF) [13,14].

In this review, we will address the open questions regarding CAF in the attempt to converge on the controversial aspects of phenotype identification and provide a glimpse into recent advances in single-cell technology-based approaches. We dissect the pathways known to drive CAF development and discuss the different functional phenotypes arising from this process. Finally, we will provide an overview of the ongoing studies investigating the therapeutic relevance of CAF, and future perspectives in the field.

## 2. The Emerging Heterogeneity within the TME Allows to Distinguish Functionally Different CAF

The term CAF generally refers to stromal cells present in the TME of solid tumors characterized by a distinct phenotype and functional and spatial patterns.

Within the tumor tissue, fibroblast activation relies on factors released by tumor cells and by tumor-infiltrating immune cells. These factors include, but are not limited to, TGF-β, platelet-derived growth factor (PDGF), and fibroblast growth factor 2 (FGF2) [13,30]. Altogether, the factors released in the TME may contribute to the acquisition of a pro-inflammatory gene signature. Indeed, these factors are responsible for driving the differentiation of stromal cells either residing in the tumor area or recruited to acquire functional and phenotypical features attributable to the CAF component, thus inducing a tumor-promoting phenotype in these cells [31].

Several researchers have attempted to identify shared criteria in order to build the basis for studying CAF starting from some common points.

The recent consensus paper published by Sahai and colleagues describes CAF as: (i) negative for epithelial, endothelial and lymphocytic markers such as E-cadherin, CD31 and CD45, respectively, and (ii) positive for markers attributable to a state of fibroblast activation, such as α-smooth muscle actin (αSMA) and fibroblast activation protein (FAP) [14]. Indeed, CAF have been identified both in vitro and in vivo based on proteins related to their activation status [13]. Over the years, these features have been complemented by the analysis of the expression of particular proteins involved in extracellular matrix (ECM) synthesis or modification, such as collagen type I and II, tenascin C, metalloproteinases (MMPs) and tissue inhibitors of metalloproteinases (TIMPs) [13]. Furthermore, CAF modify the TME through the release of factors such as TGF-β, as well as PDGF and FGFs directly affecting the differentiation of immune and stromal progenitor cells recruited in the tumor area (Figure 1B).

However, one major obstacle in the identification of a pan-marker for activated fibroblasts is represented by the marked heterogeneity of fibroblast phenotypes across different types of tumors. Indeed, to date, there are no biomarkers exclusively expressed by CAF, but rather molecules that are indicative of a change of functional state [14]. The consensus paper by Sahai and colleagues defines CAF by the combined expression of TGF-β and αSMA [14,32].

In the last decade, innovative approaches such as single-cell technology have made it possible to bring greater clarity on the complexity of CAF. Indeed, the transcriptional expression profile of CAF at a single-cell resolution has highlighted the great heterogeneity present within CAF, and to also acknowledge the uniqueness of each tumor type in this context [33,34,35]. A comparison between breast and pancreatic cancer [36] highlighted the presence of activated fibroblasts with a distinct phenotypic profile, and partially overlapping with that of normal fibroblasts [37]. Evidence of stromal cell variability in cancer emerged from a comparison of the transcriptomic profile of CAF from breast cancer with that of healthy controls via single-cell RNA sequencing (scRNAseq), whereby the presence of four distinct populations of fibroblasts inside were identified in the lymph node of the patients characterized by different metastatization ability [38]. Another study reported the presence of two distinct populations of CAF in 3D cultures of pancreatic ductal adenocarcinoma, one with myofibroblast-like morphology named myCAF, and the other one characterized by the expression of inflammatory cytokines, named iCAF [39]. Specifically, the authors reported the presence of two subsets of CAF characterized by either a myofibroblastic or an inflammatory profile, whose functional characterization correlated with the patient’s clinical response. In both subsets, the differences observed were attributable to differential modulation of TGF-β signaling [39]. Interestingly, it has been reported the possible transition between myCAF and iCAF phenotypes in a cell contact fashion [39], highlighting the plasticity of CAF. Indeed, while the differentiation of myCAF is contact-dependent, that of iCAF may occur in absence of contact with cancer cells. Similarly, functional heterogeneity was also reported in non-small-cell lung cancer, where three CAF subpopulations that differentially expressed hepatocyte growth factor (HGF) and fibroblast growth factor 7 (FGF7), were reported. The first subset characterized by high HGF expression and variable FGF7 exerted a pro-tumor effect suppressing the immune response. The second subset highly expressed FGF7 and was moderately protective of cancers, by moderately suppressing the immune system, while the last subset, presenting low level of FGF7 and HGF, scarcely suppressed the immune system [40].

At the moment, three distinguishable fibroblast subpopulations inside the TME are reported which can be principally classified as: myCAF, iCAF and antigen presenting CAF (apCAF) (Figure 1D).

### 2.1. MyCAF and Their Role in ECM Remodelling

myCAF represent the major CAF subset present in many tumors, and these cells have been found close to the tumor mass where they perform opposite functions [41,42]. Cytoskeleton proteins such as αSMA and vimentin (Vim) are considered specific to determine myCAF in combination with TGF-β expression and ECM-related proteins. On one side, factors released by myCAF generate a stiff TME, thus favoring, for example, tumor metastasis and chemoresistance [43,44,45]. On the other hand, TGF-β can activate multiple pathways leading to CAF activation [46,47]; for example, it can trigger the activation of sonic hedgehog (SHH) signaling in CAF, thus increasing ECM deposition [48]. Indeed, activated fibroblasts release various types of collagens, laminins, fibronectins, proteoglycans, periostins, and tenascin C that favor tumor progression and metastasis [49]. Moreover, myCAF have been shown to produce proteases, such as different members of the MMPs [47,48] that digest the ECM, thus facilitating invasion of tumor cells into the nearby tissue. For example, stromelysin 1 is produced by activated fibroblasts and acts by cleaving E-cadherin and consequently promoting a premalignant phenotype in mammary epithelial cells, characterized by EMT and invasiveness [50,51]. Furthermore, myCAF are able to guide the direct invasion of cancer cells by means of a CAF-cancer cell co-migration mechanism [52], thus suggesting also a role for CAF in governing metastasis.

### 2.2. iCAF Are Responsible for the Creation and Maintenance of an Inflammatory TME

iCAF are inflammatory fibroblasts distinguished from other subtypes by their expression of the C-X-C motif chemokine 12 (CXCL-12) and interleukin-6 (IL-6) and of the inflammatory macrophage identification marker lymphocyte antigen 6 (Ly6C) [39,53]. Conversely to myCAF that are located close to the tumor mass, iCAF are spatially located at the periphery of the tumor, and their differentiation is induced by interleukin-1 (IL-1) released from nearby macrophage populations (Figure 1C) [39,53]. iCAF, in turn, can release leukemia inhibitory factor (LIF), which is responsible for maintaining an inflammatory state [53] and inducing the recruitment of tumor-associated macrophages (TAMs) [54].

TGF-β1 stimulation was shown also to indirectly induce iCAF in esophageal squamous cell carcinoma (ESCC). Indeed, TGF-β1 stimulates tumor cells to release the chemokine (C-X-C motif) ligand 1 (CXCL-1), which, in turn, has been observed to correlate to iCAF formation [55]. Conversely, others reported the capacity of TGF-β to inhibit Interleukin 1 receptor-like 1 (IL1-R1) expression, thus antagonizing interleukin 1 alpha (IL1-α) responses and blocking iCAF generation [56].

A bioinformatic deconvoluted analysis was shown to discern CAF subtypes providing new information on CAF phenotype. Indeed, in this study, the authors identified several genes associated with myCAF and iCAF in bladder carcinoma (BLCA), squamous cell cancer in the head and neck region (HNSC), and lung adenocarcinoma (LUAD), including 4 overlapping genes between myCAF and iCAF. Notably, gene ontology analysis reported that genes co-expressed in both myCAF and iCAF were involved in the formation of ECM [57]. Furthermore, in this study, an elevated presence of myCAF correlated to poor prognosis in different types of cancer [57]. The poor prognostic value of myCAF has been reported also by others [58,59], and related to the ability of myCAF to decrease the sensitivity to specific antineoplastic drugs, therefore worsening disease prognosis. On the contrary, the role of iCAF is still unclear, as their presence does not determine a significant difference in drug sensitivity, but rather has been associated with a better prognosis regardless of their number [57].

### 2.3. An Obscure Subset of CAF: Antigen Presenting CAF (apCAF)

In addition to myCAF and iCAF, another population of CAF has been identified and termed antigen-presenting CAF (apCAF) [60]. These cells express major histocompatibility complex II (MHC-II) and CD74, lack the canonical co-stimulatory molecules, and release prostaglandin E_2_ (PGE2). On one hand, the concomitant lack of costimulatory molecules and release of PGE2 was shown to enhance the ability of apCAF to promote the expansion of CD4^+^ regulatory T cells (Tregs) [60]. On the other hand, apCAF were shown to activate CD4 and CD8 T lymphocytes present in the tumor tissue in an antigen-specific manner, thus enriching their phenotype with diverse immunomodulatory functions [60,61]. The expression of programmed death-ligand 2 (PDL2) and Fas ligand (FasL) on these cells allows them to inhibit the cytotoxic activity of CD8 lymphocytes. Indeed, the use of monoclonal antibodies to neutralize the in vivo interaction between CD8 and PDL2 or FasL expressed on CAF was shown to restore the cytotoxic action of T lymphocytes [62].

## 3. The Heterogeneity of CAF Is Driven by the TME through the Modulation of Different Pathways and Their Metabolism

### 3.1. CAF Plasticity Is Influenced by Different Signaling Pathways

A recent study on CAF subpopulations not only identified different types, but also discovered the capacity of these cells to switch between one activation state to another, exploiting the crosstalk among different pathways.

Indeed, it should be pointed out that the process of differentiation toward CAF is reversible. In fact, in pancreatic ductal adenocarcinoma [53], interleukin 1 beta (IL-1β) secretion from immune cells has emerged to be a potential initiator of nuclear factor kappa-light-chain-enhancer of activated B cells (NF-kB) signaling in fibroblasts, instructing them to produce a pro-tumorigenic secretome [31]. The activation of the interleukin 1/janus kinase/signal transducer and activator of transcription (IL-1/JAK-STAT) pathway has been reported to counteract the differentiation towards myCAF triggered by TGF-β, thus inducing CAF to assume a profile that is not only transcriptional, but more importantly functionally different and similar to iCAF.

The balance between these two pathways, regulated by the stimuli present in the microenvironment, was shown indeed to drive the differentiation of fibroblasts into either iCAF or myCAF, respectively [53].

Furthermore, the SHH pathway was shown to correlate with the spatial distribution of the different CAF subtypes. In fact, SHH pathway activation is higher in myCAF than in iCAF [63], and this may be explained by the proximal localization of myCAF to hedgehog ligands (i.e., TGF-β) that are expressed by the tumor and regulate myCAF differentiation. Notably, short-term SHH pathway inhibition results in myCAF depletion and iCAF enrichment, altering fibroblast composition toward a more fibro-inflammatory stroma [63]. On the other side, a study on pancreatic ductal adenocarcinoma (PDAC) reported how the presence of myCAF within the TME may counteract disease progression, as their depletion led to increased tumor invasion, decreased protective fibrosis and enhancement of important features related to tumor evolution, such as EMT, hypoxia and stemness [64]. The association with paracrine signaling between CAF and tumor cells has been shown to induce a partial EMT program in tumor cells, thus potentially stimulating tumor invasiveness. These findings are consistent with the notion that expression changes in fibroblasts may drive the acquisition of invasive features by epithelial tumor cells [65].

Besides TGF-β, another important cellular mechanism related to fibroblast commitment involves the wingless and Int-1 (Wnt) pathway [66]. Modification of the tumor–stroma ratio by altering the positive signaling loop of Wnt could induce the formation of iCAF locally, subsequently stimulating tumor invasiveness [66].

It must be considered that different CAF subpopulations alter the tumor itself and, as a matter of fact, co-cultures of mouse colorectal tumor organoids with iCAF, but not with myCAF, revealed a pronounced induction of markers of invasiveness (Zinc finger E-box-binding homeobox 1 (Zeb1) and Vim)) [67]. A different 3D matrix model characterized by high matrix stiffness was shown to favor Wnt-induced myCAF differentiation. Of interest, activation of fibroblasts by either FGF2 or TGF-β was shown to induce different tumor-promoting CAF populations, a process regulated via the transcription factor ETS Variant Transcription Factor 1 (ETV1). FGF2-activated CAF increased inflammation by recruiting macrophages, while TGF-β-activated CAF promoted invasion and EMT in squamous cell carcinoma cells [68].

### 3.2. CAF Heterogeneity Is Dependent on Metabolic Variations

In addition to secreted factors and ECM-related protein synthesis, activated fibroblasts, in contrast to healthy fibroblasts, present some modifications in the basal metabolism as well as in mitochondrial functionality [69]. Indeed, the desmoplastic process that converts fibroblasts and other precursors to myCAF is also responsible for the generation of a hypoxic microenvironment [70,71]. Hypoxia, in turn, is responsible for the maintenance of oxidative stress. Healthy fibroblasts present a metabolism based on oxidative phosphorylation in a quiescent state and gradually acquire a glycolytic metabolism as a result of the high levels of reactive oxygen species (ROS) and factors released by tumor and infiltrating immune cells, consequently altering physiological tissue homeostasis [69].

Besides glycolysis and glutamine metabolism, there is increasing evidence that lipid modification in CAF can directly or indirectly impact tumor progression. On one hand, CAF can synthesize and transfer lipids to neighboring cells affecting their behavior. For example, CAF from colorectal cancer cells (CRC) undergo lipidomic reprogramming that enhances their migration through lipid metabolites crosstalk [72]. Accordingly, exogenous lipid intake can support the rapid proliferation of tumor cells due to the upregulation of genes associated with EMT [73]. Furthermore, lipids released by CAF, including long-chain and unsaturated fatty acids, phospholipids, diacylglycerols and cholesterol esters, can change plasma membrane viscosity of CRC promoting malignancy by increasing glucose uptake and, subsequently, glycolysis [74]. Moreover, lipids derived from CAF have been shown to increase membrane fluidity also in breast cancer cells, that, among other features, can be viewed as an overall promotion of tumor progression and invasion [75]. On the other hand, exogenous lipids may indirectly increase malignancy by creating an immunosuppressive environment [76]. Therefore, lipids, such as lysophosphatidic acid and sphingosine-1-phosphate (SPP), have been identified as important paracrine regulators that modulate the homing of immune cells to the TME [77]. However, this effect also extends to the regulation of neoangiogenesis. Indeed, SPP has been shown to stimulate members of the endothelial differentiation gene (Edg) family, which, in turn, modulates blood vessel formation and maturation. This might suggest a different way through which CAFs may impact the TME, not limited to the modulation of the ECM or immune response, but by targeting neoangiogenesis [78].

Moreover, CAF also activate metabolic pathways to buffer toxic metabolites produced by cancer cells, such as lactate, thus sustaining tumor growth [79]. For example, lysophosphatidic acid (LPA) secreted from ovarian cancer cells was shown to induce transcription of hypoxia-inducible factor 1-alpha (HIF1α) by binding its receptor LPAR on activated fibroblasts, inducing these cells to gradually switch their metabolism toward glycolysis [80]. The lactate released as waste product during this process increased the oxidative stress within the TME, concurrently triggering the activation of autophagy [80,81]. This mechanism may protect activated fibroblasts under oxidative stress, as shown by increased the LC3-phosphatidylethanolamine conjugate (LC3-II) levels, therefore avoiding cell cycle arrest in S phase [82]. The involvement of autophagy in fibroblast metabolism has been demonstrated by the loss of proliferative capacity of CAF upon inhibition of autophagic flux [82]. Notably, factors released in the microenvironment, such as LIF, have shown to induce modifications, which may also support fibroblast activation and maintenance [83].

Among the different effects exerted by CAF, we will focus in particular on the aspect of interaction with the immune system, an aspect that is highly debated today, given the importance that immunotherapy has reached in the last decade, and deepen in paragraph 4.

## 4. CAF Interactions with Immune Cells of the TME

The TME is enriched by immunosuppressive cells, whose role in invasion and metastasis of tumors has been acknowledged. In this context, numerous studies have investigated the role of the different CAF subtypes in the generation of an immunosuppressive TME, by acting on T lymphocytes, NK cells, myeloid-derived suppressor cells (MDSC), and TAMs, among others, consequently favoring tumor growth [44,84,85].

In high-grade serous ovarian carcinoma, four subpopulations of CAF, defined as CAF-S1 to S4, have been identified and shown to differentially influence the immune environment. In particular, the CAF-S1 subtype increases attraction, survival, and differentiation of CD25^+^FoxP3 Tregs via CXCL12β [86]. Similar findings were reported in breast cancer where different subsets of CAF were also identified, and where CAF-S1 enhanced Treg cell polarization through the expression of the adenosine ectonucleotidase CD73 and the inhibitory molecules B7 Homolog 3 (B7H3) and dipeptidyl peptidase-4 (DPP4) [87,88,89]. On the other hand, molecules such as PDL2, junctional adhesion molecule B (JAM2) and OX40 ligand (OX40L) have been shown to stimulate interactions between CAF and T lymphocytes, reducing T lymphocyte activation and fostering the generation of an immune-supportive TME [87,88,89].

CAF can also modulate NK cell activity. Indeed, CAF have been shown to negatively interfere with the expression of natural cytotoxicity receptor (NCR), NKp30 and NKp44, as well as with the expression of the activation receptor DNAX Accessory Molecule-1 (DNAM1) on NK cells in a contact-dependent and -independent (via PGE2) fashion [90]. However, not all CAF act through PGE2. CAF isolated from endometrial cancer have the ability to reduce NK cell activation by downregulating the ligand of DNAM1, the poliovirus receptor PVR/CD155, on their surface [91]. Others reported the ability of CAF to interfere with NK killing through the secretion of MMPs that, in turn, influences MHC-I-related protein A and B (MICA-B) expression by melanoma cells, thus counteracting NK cytotoxic activity [92].

CAF influence the TME also by recruiting MDSCs. Yang and colleagues reported on the ability of STAT3-CCL-2 signaling in CAF to promote intrahepatic choloangiocarcinoma by triggering MDSC recruitment [93]. Furthermore, iCAF release cytokines, including IL-6 and SDF1, enriched by a strong inflammatory and chemotactic action, that do not only recruit monocytes, but also induce their differentiation towards MDSCs, thus stimulating tumor progression in hepatocellular carcinoma [94]. A study performed on colorectal cancer (CRC) reported the ability of FAP^high^ CAF to stimulate MDSC recruitment via (C-C motif) ligand 2 (CCL-2) [95]. Another pathway important for MDSC differentiation is the IL6/STAT3 pathway. Indeed, IL-6 released by CAF has been shown to induce the phosphorylation of STAT3 on myeloid cells, thus stimulating their differentiation into regulatory dendritic cells [96].

Other important players in the generation of an immune-evasive TME are macrophages. CAF are able to recruit large amounts of monocytes from different body districts and induce them to differentiate towards immunes suppressive TAMs [97] and, via increased interleukin 10 (IL-10) and interleukin 33 (IL-33) and decreased interleukin 12 (IL-12), suggesting a process by which CAF trigger macrophages type 1/type 2 (M1/M2) polarization [98,99]. Furthermore, it has been reported that CAF, through the release of stromal cell-derived factor 1 (SDF1), are able to promote the presence of anti-inflammatory macrophages triggering the progression of prostate cancer [100].

## 5. Therapeutic Targeting of CAF

To date, CAF represent a topical therapeutic target considering their impact on TME. However, there are still some difficulties such as the lack of specific biomarkers, as well as the ability of these to transdifferentiate depending on the stimuli present in the microenvironment [14,64,101]. In Table 1, we summarize the current clinical trials and pre-clinical studies that target different aspects of CAF and are described in this section.

The first clinical trials involving the direct targeting of the pathway of SHH-SMO have not demonstrated therapeutic efficacy and have even led to undesired results. In fact, even if a phase I clinical trial using the SHH pathway inhibitor IPI-926 combined with gemcitabine reported good tolerability, the phase II trial instead reduced the general overall survival of the patients in comparison to the placebo counterpart (NCT01130142). The use of the same drug in combination with the FOLFIRINOX was stopped early when a separate phase II trial of IPI-926 plus gemcitabine indicated detrimental effects of this combination [102]. As a matter of fact, the addition of PEGPH20 did not improve patient overall survival (OS) or progression-free survival (PFS), thus not supporting the continuation of the clinical trial [103]. Others reported the impact of vismodegib, aSHH antagonist, together with gemcitabine for the treatment of patients affected by pancreatic cancer. The addition of vismodegib to gemcitabine did not improve overall response rate, PFS, or OS in patients with metastatic pancreatic cancer [104]. Positive results were obtained from the Val-boroPro phase II clinical trial targeting FAP protein in patients affected by metastatic colorectal cancer. Although minimal clinical effects were seen, this trial paved the route for the use of drugs targeting the CAF component as adjuvant therapy [105], for example, in combination with Docetaxel (NCT00243204) for the treatment of non-small-cell lung cancer (NSCLC), or in combination with the monoclonal antibody Pembrolizumab to trigger the immune response (NCT04171219). In addition, an ongoing phase II clinical trial in pancreatic cancer patients [106] aims to re-awaken T-cell response by targeting FAP positive CAF through inhibition of the CXCL12-CXCR4 axis with the inhibitor motixafortide (BL-8040) in combination with anti PDL-1 antibody.

Conversely, some preclinical studies that target the CAF component reported modest benefits. For instance, targeting FAP+ CAF was shown to induce a modification of the ECM and increase permeability to chemotherapeutic drugs [107,108]. CAR-T therapy targeted against FAP+ cells was shown to delay tumor growth and improve survival, and induce a modest reduction in chemoresistance [107,109,110]. Also, the application of drugs conjugated to antibodies targeted against FAPs resulted in a novel and potent strategy for cancer treatment [111]. Administration of a DNA-based FAP vaccine enabled killing of CAF by tumor-infiltrating CD8, thus facilitating ECM modification, ultimately leading to improved efficacy of chemotherapeutics in multi-drug resistant murine colon and breast carcinoma [112]. Similar findings were obtained when a FAP vaccine was employed to stimulate the immune system against CAF, enabling CD8^+^ infiltration coupled to a decrease in macrophage infiltration in different pre-clinical tumor models [113].

These observations are corroborated also by other studies reporting the increased infiltration of CD8-responsive lymphocytes coupled to a rapid, hypoxic necrosis upon CAF depletion in Lewis lung carcinoma and PDAC models [123]. Gunderson and colleagues obtained similar results in a PDAC model, but, at the same time, the authors reported that FAP+ CAF targeting combined with ionizing therapy, while enabling anti-tumor T-cell infiltration and function, did not result in sufficient tumor clearance to extend animal survival [114]. However, a drawback of these therapies is related to the low specificity of FAP for CAF. Indeed, FAP is expressed also on other cells, such as bone marrow-derived MSC or skeletal muscle cells, consequently representing an unwanted target for the therapy itself, and eventually inducing cachexia [115].

Özdemir and colleagues observed that ablation of αSMA^+^ CAF reduced the survival of PDAC-bearing mice due to increased presence of Treg cells, which, in turn, impaired immunosurveillance [64]. Indeed, concomitant administration of anti-CTLA4 antibodies improved the course of the disease by enhancing survival through a restoration of an immune response. In the same study, the authors also reported on how the depletion of the αSMA^+^ population may lead to the acquisition of an EMT program, thus accelerating tumor progression. Other studies have been pursued in the attempt to re-educate CAF and therefore to target the signaling or factors responsible for fibroblast commitment to the activated status. Some strategies have attempted to use vitamin D and vitamin A to switch off CAF activity and transform the cell from pro-tumorigenic to quiescent cells in PDAC and colon cancer models [116,117,124]. Indeed, Sherman and colleagues reported the ability of calcipotriol, a synthetic form of calcitriol and a ligand of the vitamin D receptor (VDR) expressed on pancreatic tumor cells, to reduce the expression of markers of fibrosis and inflammation [116]. Similar results were observed in colon cancer, where Ferrer-Mayorga and colleagues reported how high VDR levels associated to improved overall survival. Indeed, the administration of calcitriol inhibits the activation of normal fibroblasts, thus preventing the acquisition of an activated CAF-like phenotype, determining also in this case an improvement of overall survival [124]. Similar findings were observed also for vitamin A. Patients suffering from PDAC usually present a deficiency of vitamin A with consequent activation of pancreatic stellate cells (PSC). The administration of trans-retinoic acid was able to counteract the activation of PSC, thus maintaining the cells in a quiescent state and inducing tumor cell apoptosis [117]. As a matter of fact, a clinical trial adopting a combinatorial strategy targeting the immune system through the use of a PD-1 inhibitor in parallel to an inhibitor of the vitamin D pathway, is now ongoing in PDAC [125].

Cytokines and chemokines released by CAF have been the target of combinatorial immunotherapeutic strategies in the attempt improve the effectiveness of the therapy itself [84,93,126]. For example, targeting CXCL12 with Pleraxifor, a C-X-C chemokine receptor type 4 (CXCR4) inhibitor, was shown to induce rapid T-cell accumulation among cancer cells and act synergistically with antibodies against Programmed death-ligand 1 (PD-L1), resulting in a strong reduction of tumor cells [84]. Alternatively, attempts have been made to target some of the main pathways activated by CAF, such as the IL6-JAK1/2-STAT pathways. In particular, oncostatin M has been shown to promote the contractility of ECM through Glycoprotein 130 -Inteleukin 6 (GP130-IL6), JAK1 and Rho-associated protein kinase (ROCK) signaling thus stimulating ECM remodeling. This effect was also observed in squamous cell carcinoma (SCC), where the same signaling pathway is responsible for promoting tumor invasiveness by exploiting ECM remodeling [101]. Therefore, targeting GP130-IL6ST/JAK1-ROCK axis enabled investigators to counteract tumor invasiveness by acting on CAF [101]. Another study that targeted the CAF surface molecule GPR77^+^ reported decreased infiltration of CAF into tumors and reduced presence of cancer stem cells, consequently increasing chemosensitivity in breast cancer patient-derived xenografts. In the same study, the authors observed that targeting IL-6 and interleukin 8 (IL-8) cytokines with specific antibodies synergistically to docetaxel administration allowed an almost complete disease remission [118]. Others reported how the blocking of the IL-6/STAT3 axis was able to reduce immune suppression and commitment to MDSCs, thus putatively enhancing the effect of immunotherapy [119,120].

In addition, blocking TGF-β signaling in conjunction to the administration of anti PD-L1 antibodies, the latter of which is of the main molecules released by CAF, facilitated the penetration of T lymphocytes into the tumor and caused an effective tumor regression [121]. Also, the administration of the drug Tranilast, that suppresses the release of TGF-β by fibroblasts, has been shown to counteract tumor progression in mouse models of C-type lectin domain family 12 member A (CLEC12A/CLL1) Lewis lung cancer and B16F1 melanoma. In this case, the authors observed that the infiltration of immunosuppressive cells, including Treg and MDSC, as well as the release of immunomodulatory molecules such as SDF1 and PGE2, was reduced by the treatment [122].

## 6. Conclusions

A central role for CAF in governing the evolution of the TME by acting on various cellular and non-cellular constituents is now acknowledged, and so is their potential therapeutic targeting, enriching the field with interesting clinical perspectives. However, what has also become increasingly evident, owing to modern methods of investigation such as single-cell analysis, is the high level of heterogeneity and complexity existing within the diseased stromal compartment of tumors. Indeed, cellular and non-cellular components of cancer are endowed with broad and, at times, opposite functions, thus further nurturing an already plastic microenvironment.

Nevertheless, most of the studies discussed here present technical limitations related to the sensitivity of the method itself. In fact, poorly expressed genes are difficult to evaluate. Similarly, the level of transcripts observed does not always reflect the amounts of proteins present in the sample. Pitfalls are also related to the algorithms of analysis and representation that sometimes generate quite artificial clusters [127], as they are not based on actual functional differences, and also lack information on spatial positioning that may provide relevant cues for cell to cell interactions.

Therefore, future efforts should focus on understanding the functional properties of individual subtypes of CAF, taking into consideration the peculiarity of each pathological context.

## Figures and Tables

**Figure 1 cancers-14-03570-f001:**
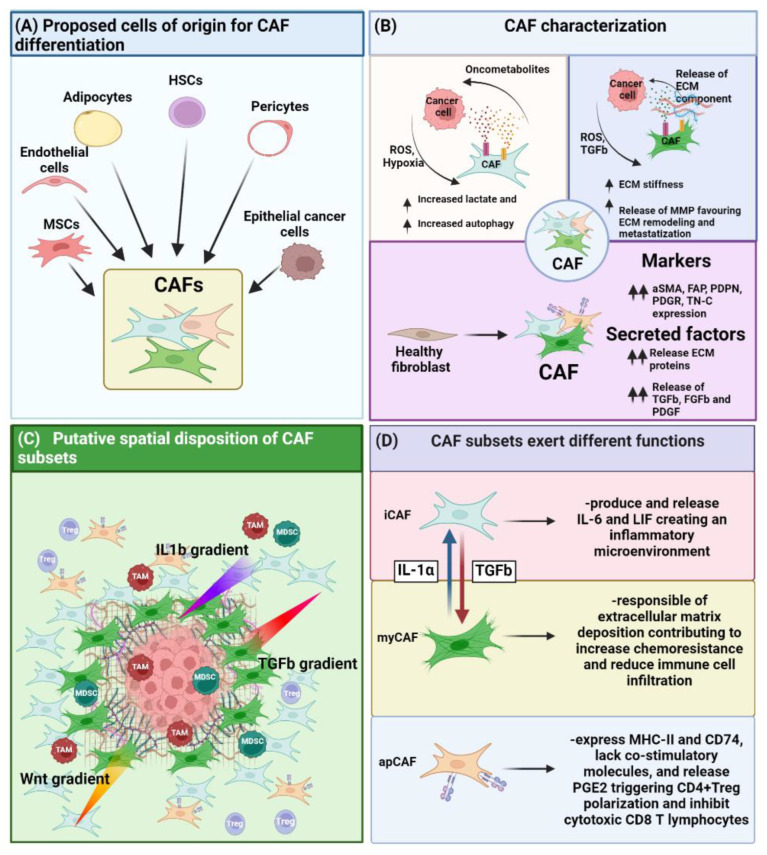
**Role of CAF on the TME**. (**A**) CAF can differentiate from various cell types such as MSC, endothelial, adipocyte, HSC, pericyte and epithelial cancer cells upon EMT. (**B**) CAF have been characterized under different point of views, ranging from their metabolic impact on the TME, their ability on altering ECM stiffness, as well as by the phenotypical and functional point of view enlisting a series of markers as well as factors they must secrete. (**C**) Different CAF subsets present a differential spatial disposition inside the TME. This effect is strictly dependent on the release by the tumor as well as by the immune cells of cytokines or growth factor, creating a gradient of differentiation triggering either the commitment towards myofibroblast-like CAF (myCAF) (TGF-β gradient), inflammatory CAF (iCAF) (IL-1β/Wnt gradient,). (**D**) Different CAF subsets exert different function.

**Table 1 cancers-14-03570-t001:** Clinical trials and preclinical studies targeting CAF.

Target	Status	Results	Treatment	Reference
SHH-SMO	Clinical trial	Toxicity in patients with pancreatic ductal adenocarcinoma	IPI-926+FOLFIRINOX	[102]
Hyaluronic acid	Clinical trial	Toxicity in patients with pancreatic ductal adenocarcinoma	PEGPH20+nab-paclitaxel	[103]
SHH antagonist	Clinical trial	Not improve overall response rate, PFS, or OS in patients with pancreatic cancer	Vismodegib+gemcitabine	[104]
FAP	Clinical trial	Minimal clinical effect in patients with metastatic colorectal cancer	Val-boroPro (Talabostat)	[105]
FAP	Clinical trial	BL-8040 increased CD8+ effector T-cell tumor infiltration, decreased myeloid-derived suppressor cells (MDSCs) and further decreased circulating regulatory T cells	Inhibitor motixafortide (BL-8040) (CXCR4 inhibitor) in combination with anti PD-1 antibody	[106]
αSMA	Preclinical study	Reduced the survival of PDAC-bearing mice due to increased presence of Treg cells and lack of effect by gemcitabine treatment. The co-administration of anti-CTLA4 reversed disease acceleration and prolonged animal survival	Transgenic mouse for aSMA+Gemcitabine +/− anti-CTLA4	[64]
CXCR4	Preclinical study	Rapid T-cell accumulation among cancer cells and act synergistically with antibodies against PD-L1, resulting in a strong reduction of tumor cells	Targeting CXCL12 with Pleraxifor (AD3100), a CXCR4 inhibitor	[84]
FAP	Preclinical study	Modification of the ECM and increase permeability to chemotherapeutic drugs	CAR-T targeting FAP	[107]
FAP	Preclinical study	β1-integrins may abrogate the invasive capabilities of pancreatic and other tumors by disrupting the FAP-directed organization of stromal ECM	β1-integrin antibody mAb13 and the α5β1- integrin blocking peptide ATN-161	[108]
FAP	Preclinical study	Inhibited the growth of multiple types of subcutaneously transplanted tumors in wild-type mice	CAR-T targeting FAP	[109]
FAP	Preclinical study	Enhanced overall antitumor activity and conferred a survival advantage in a systemic A549 tumor model	CAR-T targeting FAP+T cells that targeted the EphA2	[110]
FAP	Preclinical study	OMTX705 treatment increased tumor infiltration by CD8+ T cells, induced complete regressions, and delayed tumor recurrence.	OMTX705 anti FAP antibody+/− chemotherapy or immunotherapy (anti PD-1)	[111]
FAP	Preclinical study	Killing of CAF by tumor-infiltrating CD8, thus facilitating ECM modification, improved efficacy of chemotherapeutics in multi-drug resistant murine colon and breast carcinoma	DNA-based FAP vaccine	[112]
FAP	Preclinical study	SynCon FAP vaccine in combination with other DNA vaccine induce better OS in prostate and breast cancer mouse model	SynCon FAP vaccine in combination with a PSMA vaccine or TERT DNA vaccines	[113]
FAP	Preclinical study	Enabled anti-tumor T-cell infiltration and function, did not result in sufficient tumor clearance to extend animal survival	UAMC-1110 a new FAP small molecule inhibitor+focal radiotherapy	[114]
Vitamin D receptor (VDR) and Vitamin A receptor (STRA6)	Preclinical study	To turn off CAF activity and transform the cell from pro-tumorigenic to quiescent cells in PDAC and colon cancer models	Vitamin D and vitamin A	[115,116]
Retinoic acid receptor (RAR)	Preclinical study	Counteract the activation of PSC thus maintaining the cells in a quiescent state and inducing tumor cell apoptosis	Trans-retinoic acid	[117]
GP130-IL6ST/JAK1-ROCK	Preclinical study	JAK1/2 silencing reduce ROCK and IL6ST activation and are useful to block invasion and metastasis	GP130-IL6ST/JAK1-ROCK silencing	[101]
GPR77+, IL-6, IL-8	Preclinical study	Targeting the CD10+GPR77+ CAF subset abolishing tumor formation and restores tumor chemosensitivity	Neutralizing anti-GPR77 antibody and IL-6 and IL-8 cytokines with specific antibodies synergistically to docetaxel administration	[118]
GP130-IL6ST/STAT3 pathway	Preclinical study	Reduce immune suppression and commitment to MDSCs, thus putatively enhancing the effect of immunotherapy	Blocking of the IL-6/STAT3 axis	[119]
IL-6, FLLL32-STAT3	Preclinical study	Abrogated pancreatic stellate cells mediated MDSC differentiation, thus improving PSC viability	IL6 blocking and FLLL32 STAT3 inhibitor	[120]
TGF-β and PD-L1	Preclinical study	Facilitated the penetration of T lymphocytes into the tumor and caused an effective tumor regression	Blocking TGF-β signaling in conjunction to the administration of anti PD-L1 antibodies	[121]
TGF-β production and release	Preclinical study	Counteract tumor progression in mouse models of CLL1 Lewis lung cancer and B16F1 melanoma. with reduced immune-suppression by tumor cells	Tranilast, suppressor of TGF-β release	[122]

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
