# Peer review of "Fight the Cancer, Hit the CAF!"

_cancers, 2022, doi:10.3390/cancers14153570_

Round 1

Reviewer 1 Report

3.2. CAF heterogeneity is dependent on metabolic variations may be revised to highlight the activated fibroblasts with the addition of description on the ROS-induced activation.

Table 1 needs the revision in terms of beta1-integrin and JAK1/2.

Author Response

AUTHOR REPLY: We thank the reviewer for the valuable comments which allow us to improve our manuscript.

We have added the required commentary in section 3.2 (lines 250-254) and revised the parts on β1 integrin and JAK1/2 (see table).

Reviewer 2 Report

This is a review of CAF / cancer associated fibroblasts that differentiate from various cells while considering the pathophysiology of cancer or cancer microenvironment. From a historical perspective, it is a comprehensive review that includes characterization with the latest shingle cells. It also mentions therapeutic attempts targeting various molecules and proteins as the characteristics and properties of various CAFs.

It is a very good review article and finds no particular problems.

Author Response

AUTHOR REPLY: We thank the reviewer for the revision and for the positive comments.

Reviewer 3 Report

11.      “Indeed, there is only one purpose, to promote its growth through immune evasion, stimulation of neo-angiogenesis, as well as also supporting the process of tumor proliferation and metastasis” (raw 63-65) – It seems like a strong statement. In fact, stromal cells can also exert antitumor effects. For example, TGFb synthesized in CAFs in the early stages of tumor development can suppress the division of cancer cells, and “apCAFs were shown to activate CD4 and CD8 T lymphocytes present in the tumor tissue in an antigen-specific manner”. I think this statement needs to be softened.

22.   “On one side, factors released by myCAFs generate a stiff TME, thus favoring for example tumor metastasis chemoresistance by the tumor (42-44).” (raws 141-143). May be the authors meant here “tumor metastasis AND chemoresistance by the tumor”?

33.    “In this work the authors identified a total of 14 and 413 differentially expressed genes between myCAF and iCAF, respectively, and revealed only 4 genes expressed by both myCAF and iCAF” (raws 171-173). This phrase is incomprehensible. Apparently, the transcriptomes of these CAFs populations were compared with the transcriptomes of some other cells when DEGs were identified. Сlarification needs to be made here.

44.     «given the importance that immunotherapy has reached in the last decade and deepen in the paragraph 3.» (raws 258-260) – May be the authors meant here «the paragraph 4»?

55.     It is necessary to carefully check the reference column in table 1, since the showed references do not correspond to the list of references (within the meaning of) and the text of paragraph 5.

66.      It is necessary to mention in chapter 5 about clinical trials for talabostat (Val-boroPro, FAP inhibitor)

Author Response

Comments and Suggestions for Authors

AUTHOR REPLY:

  1. “Indeed, there is only one purpose, to promote its growth through immune evasion, stimulation of neo-angiogenesis, as well as also supporting the process of tumor proliferation and metastasis” (raw 63-65) – It seems like a strong statement. In fact, stromal cells can also exert antitumor effects. For example, TGFb synthesized in CAFs in the early stages of tumor development can suppress the division of cancer cells, and “apCAFs were shown to activate CD4 and CD8 T lymphocytes present in the tumor tissue in an antigen-specific manner”. I think this statement needs to be softened.

AUTHOR REPLY: As suggested, we modified the sentence and softened the statement (line 64-67).

  1. “On one side, factors released by myCAFs generate a stiff TME, thus favoring for example tumor metastasis chemoresistance by the tumor (42-44).” (raws 141-143). May be the authors meant here “tumor metastasis AND chemoresistance by the tumor”?

AUTHOR REPLY: We apologize for the error, which we have now corrected (line 146).

  1. “In this work the authors identified a total of 14 and 413 differentially expressed genes between myCAF and iCAF, respectively, and revealed only 4 genes expressed by both myCAF and iCAF” (raws 171-173). This phrase is incomprehensible. Apparently, the transcriptomes of these CAFs populations were compared with the transcriptomes of some other cells when DEGs were identified. Сlarification needs to be made here.

AUTHOR REPLY: We have modified according to the reviewer’s suggestion and we attempted to clarify the concept (lines 174-178).

  1. «given the importance that immunotherapy has reached in the last decade and deepen in the paragraph 3.» (raws 258-260) – May be the authors meant here «the paragraph 4»?

AUTHOR REPLY: We thank the reviewer for pointing out the error which we have now corrected (line 296).

  1. It is necessary to carefully check the reference column in table 1, since the showed references do not correspond to the list of references (within the meaning of) and the text of paragraph 5.

AUTHOR REPLY: We have completely revised the table by assessing the correspondence of the individual references to the indicated content

  1. It is necessary to mention in chapter 5 about clinical trials for talabostat (Val-boroPro, FAP inhibitor)

AUTHOR REPLY: We have added the indicated trials (lines 367-373) and implemented the table.

Reviewer 4 Report

This is an interesting review that sets the stage for further research on CAF.

Conceptually, the work need to be broader and discuss also the role of lipids in the process.  For example:

Regulation of Tumor Immune Microenvironment by Sphingolipids and Lysophosphatidic Acid. Curr Drug Targets. 2022;23(6):559-573. doi: 10.2174/1389450122666211208111833. PMID: 34879798. and

 Edg-1, the G protein-coupled receptor for sphingosine-1-phosphate, is essential for vascular maturation. J Clin Invest. 2000 Oct;106(8):951-61. doi: 10.1172/JCI10905. PMID: 11032855; PMCID: PMC314347.

In addition, the english and use of abbv/formatting could be improved.

Please define TME in the abstract.

Line 55 derived/derived seems redundant. Find better wording.

Line 56 define MSC.Line67  cancer 66 associated fibroblastS vs CAF

Line 68. Use CAF here and thereafter, since the S is embeded now.

Line 299  iCAF/ release  should not have a slash. Same issue in other lines.

The table in line 335 formatting can be improved. Remove the extra squares. Concordant use of capitalization.....

Author Response

Comments and Suggestions for Authors

This is an interesting review that sets the stage for further research on CAF.

Conceptually, the work need to be broader and discuss also the role of lipids in the process.  For example:

Regulation of Tumor Immune Microenvironment by Sphingolipids and Lysophosphatidic Acid. Curr Drug Targets. 2022;23(6):559-573. doi: 10.2174/1389450122666211208111833. PMID: 34879798. and

 Edg-1, the G protein-coupled receptor for sphingosine-1-phosphate, is essential for vascular maturation. J Clin Invest. 2000 Oct;106(8):951-61. doi: 10.1172/JCI10905. PMID: 11032855; PMCID: PMC314347.

AUTHOR REPLY: We thank the reviewer for the very useful comment that has allowed us to improve our review. We have added a part on the impact of lipids in the TME in section 3.2 (lines 261-278).

In addition, the english and use of abbv/formatting could be improved.

AUTHOR REPLY: We thank the reviewer for the comments made and apologize for any errors. Below is the timely response for each comment.

Please define TME in the abstract.

AUTHOR REPLY: We modified as indicated by the reviewer (line 13)

Line 55 derived/derived seems redundant. Find better wording.

AUTHOR REPLY: We modified as suggested by the reviewer (line 59)

Line 56 define MSC.

AUTHOR REPLY: We implemented as suggested by the reviewer (line 57)

Line 67 cancer associated fibroblastS vs CAF

AUTHOR REPLY: We implemented as suggested by the reviewer directly in the abstract (line 23) and the first time it is cited in the text (line 70).

Line 68. Use CAF here and thereafter, since the S is embeded now.

AUTHOR REPLY: We have edited throughout the text as suggested by the reviewer

Line 299 iCAF/ release should not have a slash. Same issue in other lines.

AUTHOR REPLY: We have edited along the entire text as suggested by the reviewer, we apologize for the error

The table in line 335 formatting can be improved. Remove the extra squares. Concordant use of capitalization.

AUTHOR REPLY: We thank the reviewer for the valuable comment. We have edited the table.

Round 2

Reviewer 1 Report

The manuscript has been improved.

Author Response

We thank the Reviewer for helping us improve our manuscript.

Reviewer 3 Report

Thanks, all my comments have been taken into account.

When re-viewing table 1, I had next question. For the first clinical trial in the table, the target is SHH SMO, but treatment is "Pegylated recombinant human hyaluronidase (PEGPH20) +FOLFIRINOX". Perhaps there is some mistake here. Otherwise, it is necessary to explain which compound from the pharmaceutical combination directly affects the target (SHH SMO).

Author Response

Dear Reviewer,

Thank you once again for your comments and we corrected the mistake in section 5 and in Table 1.

We hope that the manuscript can now be considered suitable for publication.

Reviewer 4 Report

Still needs grammar/abbreviation improvement. For example in the abstract

What is TME

Im-mune (sic)

Author Response

Dear Reviewer,

We have edited and corrected some typos throughout the text. 

Thank you for pointing these out.